# Spatial-Temporal Pattern and Driving Forces of Fractional Vegetation Coverage in Xiong'an New Area of China from 2005 to 2019

**Zhiqing Huang** [1,2,3], **Haitao Qiu** [4], **Yonggang Cao** [5], **Adu Gong** [1,2,3,*] and **Jiaxiang Wang** [6]

1   Beijing Key Laboratory of Environmental Remote Sensing and Digital City, Department of Geographic Science, Beijing Normal University, Beijing 100875, China; huangzhiqing@mail.bnu.edu.cn
2   State Key Laboratory of Remote Sensing Science, Faculty of Geographical Science, Beijing Normal University, Beijing 100875, China
3   Beijing Engineering Research Center for Global Land Remote Sensing Products, Faculty of Geographical Science, Beijing Normal University, Beijing 100875, China
4   Aerospace Long-March International Trade Co., Ltd., Beijing 100054, China; haitaoqiu@alitcn.com
5   Weihai Municipal Water Resources Affairs Service Center, Weihai 264200, China
6   College of Earth Science, Chengdu University of Technology, Chengdu 610059, China; wangjiaxiang@stu.cdut.edu.cn
*   Correspondence: gad@bnu.edu.cn

**Abstract:** The Xiong'an New Area was officially established in 2018 to construct a new, intelligent, and efficient urban area to alleviate Beijing's non-capital functions. Using Landsat satellite images, we employed the dimidiate pixel model, band operation, and transition matrix to analyze the temporal and spatial variations in FVC (Fractional Vegetation Coverage) within the Xiong'an New Area in 2005, 2013, and 2019, respectively. Urbanization rate, precipitation, temperature, and population were considered potential driving forces, which we analyzed using grey relational analysis and linear regression to explore the correlation between FVC and these factors. The findings are as follows: from 2005 to 2019, overall improvement and significant degradation have been observed. In Baiyangdian, a part of the national key ecological area, water bodies and FVC have increased. Grey relational analysis revealed that precipitation had the highest grey relational value of 0.76. The average correlation among natural factors was 0.67, while that among human factors was 0.60. Generally, the Xiong'an New Area vegetation exhibited instability, while Baiyangdian demonstrated relatively stable FVC. Grey relational analysis indicates a strong potential for social and economic development in the Xiong'an New Area.

**Keywords:** Xiong'an New Area; fractional vegetation coverage; spatial and temporal variation; dimidiate pixel model; grey relational analysis

## 1. Introduction

FVC (Fractional Vegetation Coverage) refers to the percentage of the vertically projected area occupied by aboveground vegetation, including leaves, stems, and branches, within a specified unit area [1]. It serves as a comprehensive and quantitative indicator that accounts for the surface condition of plant communities [2]. Moreover, FVC is an essential parameter for describing vegetation communities and ecosystems [3], providing insights into the state of the regional ecological environment to a certain extent [4]. Urbanization reflects the urban and regional socio-economic development level, which the urbanization rate can quantify over a specific period.

Remote sensing images are essential for vegetation cover research, offering valuable data for various land use studies. Landsat (TM/ETM+/OLI) has been a primary data source due to its high spatial resolution, though its temporal resolution lags behind that of medium-resolution sources like MODIS [5–7]. Researchers often enhance the spatial

and temporal resolution of remote sensing images through data fusion methods, such as STARFM and ESTARFM [8]. Some scholars have improved these methods by incorporating surface heterogeneity information, resulting in more efficient and robust algorithms with enriched fusion results [9]. However, despite breaking the barrier between medium-resolution remote sensing images, fused data may not be well-suited for areas with rapid or slight changes in land use [8]. In specific scenarios, Landsat data acquired in relatively flat areas with favorable conditions provides reliable and unaltered data for research, obviating the need for data fusion [10]. On the other hand, Sentinel-2 data, with a 10 m resolution and higher spectral resolution in the red-edge area of plants, is commonly used for vegetation analysis but may be more suitable for single-image analysis rather than long-term studies [11]. For spatiotemporal vegetation studies, scholars can directly analyze SPOT/VEGETATION NDVI data produced by the SPOT satellite [12]. In recent years, the GEE (Google Earth Engine) platform has gained popularity among scholars for remote sensing analysis. This cloud-based platform offers technical support for large-scale research and data analysis, including applications like vegetation index analysis, vegetation cover assessment, and so on [13–15].

Researchers have conducted spatial and temporal analysis on vegetation in recent years using remote sensing data. Some scholars have utilized the coefficient of variation model and trend analysis methods to investigate China's human-influenced pattern of vegetation coverage [16]. Hussain applied the maximum likelihood method into a classification and through thorough analysis on major crops in Pakistan from 1984 to 2020, found information on the relationship between crops and climate factors [17]. The Hurst index, Mann-Kendall significance test, Theil–Sen Median method, gravity center shift analysis, and transition matrix are also common strategies for spatial-temporal analysis for remote sensing data. Mao used the former three methods to investigate the LAI (Leaf Area Index) fluctuation rule and dynamic change from 2003 to 2020 in the Guangdong province, China [18]. And the result indicated there is a positive trend for vegetation. Zhang found that the vegetation coverage's spatial distribution in the Beijing–Tianjin–Hebei region of China is unbalanced by a gravity center shift analysis, and the overall trend is westward [19]. Scholars also used a cellular automata Markov chain (CA-Markov chain) to reveal the variation of LUCC (Land-Use and Land-Cover Change) [20]. In addition to various probabilistic models and mathematical statistical methods, machine learning methods have become prevalent in spatial-temporal processing, among which RF (Random Forest), SVM (Support Vector Machine), and kinds of neural networks are adopted most often, and RF typically yields more accurate results compared to alternative methods [21,22]. In summary, the aforementioned methods mostly analyze and study the overall trends and characteristics of vegetation from a macroscopic perspective. While this approach effectively captures the general features of changes, it lacks the ability to specifically reveal the direction of changes. On the other hand, methods like land-use transition matrices provide a more intuitive representation.

Various methods are commonly employed for driving force analysis, including factor analysis, principal component analysis, regression analysis, Mann-Kendall trend method, geographic detector, and so on. Many scholars have utilized these methods to identify the factors most closely associated with vegetation change, such as climate variables and human activities (e.g., precipitation, temperature, and population density) [23–25]. These factors can be considered potential driving forces behind vegetation change. However, factor analysis, principal component analysis, and regression analysis primarily analyze attribution from the perspective of mathematical statistics [26], often lacking sufficient consideration of the temporal dimension. Geo-detectors focus on detecting factor differentiations from a spatial perspective [27]. The Mann-Kendall trend method is well-suited for analyzing time series data with continuous increasing or decreasing trends, namely monotonic trends [28]. Another approach, the grey correlation degree analysis method, is proposed based on the grey system theory and is suitable for analyzing data series with time trends. It assesses the correlation between data by examining the reasonable degree of

data changes [29]. Although widely used in statistical analysis, its application in driving force analysis remains relatively limited.

However, despite numerous studies on vegetation coverage in different locations, there is still a lack of targeted spatial-temporal feature analysis for typical cities and regions within the Beijing–Tianjin–Hebei area, especially in terms of recent research within the past five years. Regarding the Xiong'an New Area, recent studies have primarily focused on factors such as net primary production (NPP), gross primary production (GPP), and land use. These studies have shown that changes in precipitation, temperature, and eCO2 contribute to variations in GPP, while local NPP tends to fluctuate and is primarily influenced by urbanization and agricultural technology levels [30,31]. Dryland, rural land, and paddy fields dominate land use in the Xiong'an New Area [32]. Nonetheless, there remains a lack of studies on the local vegetation spanning a time horizon of more than ten years. As a mirror to the ecosystem, vegetation is highly sensitive to climate and environmental changes, evolving rapidly. Also, vegetation management is a substantial part of the new area's planning. Therefore, it is essential to continue monitoring its changing patterns and underlying causes. Xiong'an is a newly planned area established by the Party Central Committee after the Shenzhen Special Economic Zone and Shanghai Pudong New Area. According to the "Hebei Xiong'an New Area Planning Outline", issued by the Hebei Provincial Committee of the Communist Party of China in 2018, the original intention behind its construction was to relieve Beijing of its non-capital functions and develop it into a new urban area characterized by green development, intelligence, efficiency, and high-quality economic growth [33]. In addition to industrial development, the area aims to maintain a favorable ecological environment and establish a sustainable development model that harmonizes green practices with high efficiency, as it encompasses Baiyangdian, a critical national ecological area.

Therefore, addressing the current issues in related research above, this study analyzes the spatial and temporal changes in vegetation coverage in the Xiong'an New Area from 2005 to 2019. Using the grey correlation degree method, it also investigates the correlation between local urbanization development level, population, temperature, precipitation, and FVC. The findings aim to provide insights for further developing a green city in the Xiong'an New Area. The remainder of this paper is organized as follows: Section 2 provides an overview of the study area and data. Section 3 presents the spatial-temporal pattern of FVC in Xiong'an from 2005 to 2019. Section 4 presents the correlation analysis between FVC and driving factors. Finally, Section 5 presents the discussion, which concludes the work and proposes recommendations for further development.

## 2. Materials and Methods

### 2.1. Study Area

Xiong'an New Area is in the central region of Beijing, Tianjin, and Baoding. It encompasses Xiong County, Rongcheng County, and Anxin County (Figure 1). The planned area covers a total of 1770 square kilometers, with longitude ranging from 115°37′52″ to 116°21′17″ and latitude ranging from 38°40′2″ to 39°12′15″. As of the seventh national population census, the permanent population of Xiong'an New Area is 1.2 million people. The urbanization rate in the new area has surpassed 40%. Xiong'an New Area experiences a temperate continental monsoon climate due to its location in the mid-latitude region. The average annual temperature is 13.17 °C, and the average annual precipitation is 495.1 mm. The average annual sunshine hours are 2308.4 h [33].

The representative vegetation in the area is temperate deciduous broad-leaved forest. The terrain gradually declines from the northwest to the southeast, with most ground elevation ranging from 5 to 26 m and the ground slope is less than 2‰. The area primarily consists of a stacked plain landform in the eastern part of the Taihang Mountains. The terrain within the area is predominantly flat, and the soil is fertile. The land is mainly covered by agricultural fields, with cultivated land accounting for 93% of the agricultural area, and forested land covering 3.4% [7].

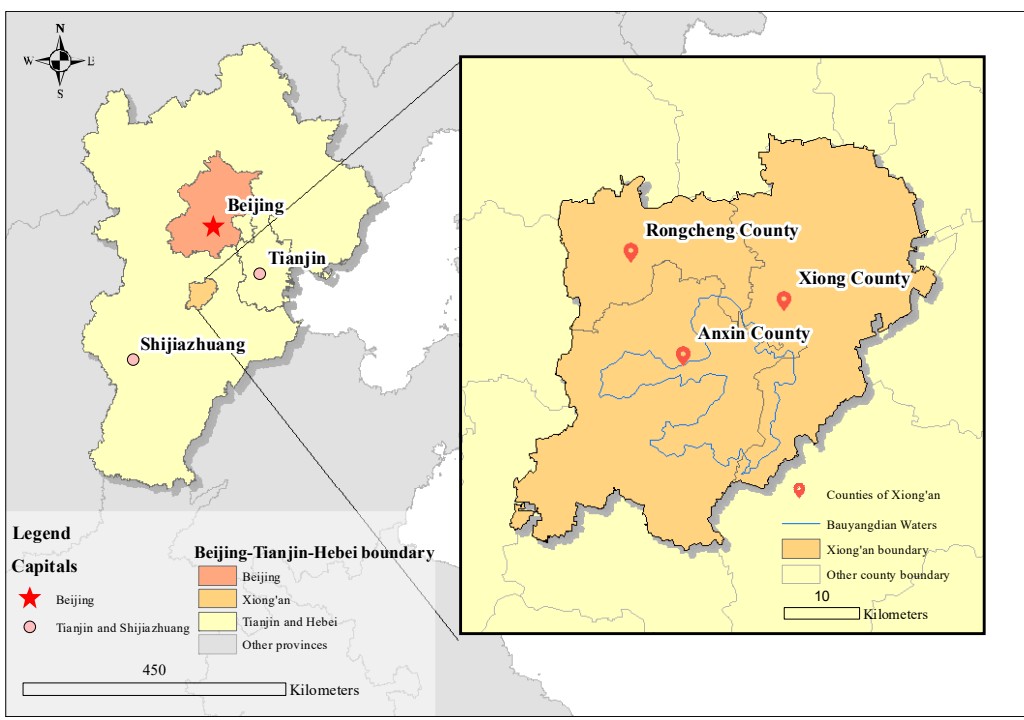

**Figure 1.** The study area.

Xiongxian, Rongcheng, Anxin Quanjing, and their surrounding regions are part of the Daqing River system in the Haihe River Basin. The area has a well-developed water system with a dense network of rivers and canals, with a river network density ranging from 0.12 to 0.23 km/km$^2$. The most prominent water body in the region is the Baiyang Lake. Baiyang Lake is the largest freshwater lake in the North China Plain, with a total area of 366 km$^2$, out of which 312 km$^2$ is located in Anxin County [34].

## 2.2. Data Sources

The remote sensing, vegetation, climate, and population data were obtained from various sources. Details on the data type and spatial and temporal resolution can be found in Table 1.

**Table 1.** Summary of the collected datasets.

| Data Type | Data Name | Time | Spatial and Temporal Resolution | Source | Format |
|---|---|---|---|---|---|
| Remote sensing | Landsat 7 ETM+ | 2005 | 30 m; daily | Resource and Environment Science and Data Center | .tiff |
| | Landsat 8 OLI-TRIS | 2013, 2019 | | United States Geological Survey | .tiff |
| Vegetation | SPOT/VEGETATION NDVI | 2005–2019 | 1 km; month | Resource and Environment Science and Data Center | .tiff |
| Climate | Precipitation; Average Temperature | 2005–2019 | Weather Stations *; daily | China Meteorological Administration | .csv |
| Population | Total population at the end of the year | 2005–2019 | County; year | National Bureau of Statistics | text |

| Data Type | Data Name | Time | Spatial and Temporal Resolution | Source | Format |
|-----------|-----------|------|--------------------------------|--------|--------|
| Boundary | Administrative divisions | 2017 | 1:1 million; year | Ministry of Natural Resources of China | .shp |
| Land cover | GLC_FCS30-1985_2020 | 2005, 2015, 2020 | 30 m; year | CASEarth | .tiff |

\* Three weather stations' observation data are used: 54,503 (39°4′ N, 115°49′ E), 54,605 (38°56′ N, 115°56′ E), and 54,636 (39°1′ N, 116°6′ E), which are located in Rongcheng County, Anxin County, and Xiong County, respectively.

### 2.2.1. Remote Sensing Data

The remote sensing data utilized in this study consists of Landsat satellite images. For the analysis of FVC spatial-temporal changes, Landsat 7 ETM+ remote sensing images from 2005 and Landsat 8 OLI-TRIS remote sensing images from 2013 and 2019 are employed. In our study, our primary focus lies on capturing the intuitive and subtle spatial-temporal variations of FVC. We achieve this through pixel-by-pixel variation maps and transition matrices. These methods are best suited to images with specific year intervals. Consequently, the spatiotemporal variations in FVC within each interval can be accurately depicted using the corresponding pixel-by-pixel transition map and a grade transition matrix of FVC, allowing for mutual interpretation and enhancing the reliability and consistency of the results. Other studies have also employed Landsat imagery and other time series data at certain interval years for long-term investigations of vegetation or other land cover dynamics. Commonly adopted time intervals for studies spanning approximately twenty years include 5-year [35] and 10-year increments [36,37], while some researchers have opted for strategies without specific intervals to obtain more macroscopic results [38]. Therefore, we selected three years' worth of images for our analysis, covering a time range of 2005–2019.

Our images are obtained from two sources: the Resource and Environment Science and Data Center (http://www.resdc.cn/) (accessed on 30 October 2021), and the official website of the United States Geological Survey (https://www.usgs.gov/) (accessed on 21 September 2021). The images are collection 1 level 1 data, which ensures their reliability and quality. The imaging time is specifically chosen during summer when vegetation growth is optimal, and the weather is clear. The resolution of the remote sensing images is 30 m, providing a detailed view of the study area.

### 2.2.2. Vegetation Data

In this study, the vegetation data is obtained from two different sources. Firstly, for the analysis of spatial-temporal changes in FVC, Landsat satellite images are used. These images are processed independently to calculate FVC. Secondly, for the grey correlation analysis between FVC and its driving factors and result verification, the SPOT/VEGETATION NDVI dataset is utilized. This dataset is sourced from the Resource and Environmental Science and Data Center. The SPOT/VEGETATION NDVI dataset provides information on the vegetation index, specifically the China Long-term Vegetation Index Dataset. The dataset is based on imaging conducted in August of each year and has a spatial resolution of 1 km, covering the same time range with the remote sensing images.

### 2.2.3. Climate Data

This study's climate data for the Xiong'an New Area include precipitation and daily average temperature. The precipitation data is derived from the 24 h cumulative precipitation recorded from 20:00 to 20:00 the following day. The daily average temperature data is also obtained. These data are from three weather stations in the Xiong'an New Area. The annual average temperature and annual cumulative precipitation are calculated based on daily data. The weather station data used in this study is sourced from the China Meteorological Science Data Sharing Service Network (https://data.cma.cn/) (accessed

on 21 November 2021), which belongs to the China Meteorological Administration, with a time range of 2005–2019.

### 2.2.4. Population Data

The population data used in this study were obtained from the Statistical Yearbook of Chinese Cities, covering 2006 to 2020 versions (each version is a record of the last whole year), published by the National Bureau of Statistics. This study's climate data for the Xiong'an New Area include precipitation and daily average temperature. By aggregating the population numbers from these counties, we obtained the total population for the Xiong'an New Area. The urbanization rate, which indicates the distribution of the permanent population between urban and rural areas and reflects the level of urbanization, was calculated by dividing the urban population by the total population. It is important to note that the urbanization rate is determined based on the permanent resident population rather than the nature of household registration, as the National Bureau of Statistics specified.

### 2.2.5. Boundary Data

The boundary data used for cartographic representation, including national boundaries, provincial boundaries, and county boundaries, were obtained from the National Basic Geographic Information Center under the Ministry of Natural Resources of China (https://www.ngcc.cn/ngcc/) (accessed on 5 August 2020). All the data were provided in vector format.

### 2.2.6. Land Cover Data

The land cover data (GLC_FCS30-1985_2020) is used to assist in analyzing the spatial distribution and changes of FVC [39]. This dataset employs a combination of change detection and dynamic updating, utilizing Landsat imagery (Landsat TM, ETM+, and OLI) to derive 29 land surface cover types. The data spans from 1985, with intervals of 5 years. It can be obtained on the data sharing service system of CASEarth (https://data.casearth.cn/) (accessed on 21 July 2023). The dataset improves upon previous datasets by reducing misclassifications and omissions. It demonstrates good spatial continuity in transitions and achieves an overall accuracy of 82.5%. Through comparison, this dataset effectively reflects the true land cover categories in the study area. For our analysis, we selected the years 2005, 2013, and 2019, which are the closest available years to 2005, 2015, and 2020, respectively. In this study, we have reclassified land cover into seven major categories: crop, grass, forest, wetland, impervious surfaces, bare land, and water bodies.

### *2.3. Methods*

The experiment followed the main technical route outlined in Figure 2. The primary data sources for the experiment consisted of remote sensing images, yearbooks, and meteorological station data. Each data source underwent specific preprocessing procedures. The calculations were conducted to analyze the spatial and temporal changes in FVC and the influencing factors. The results were then analyzed in conjunction with the actual situation. The grey correlation degree method examined the relationship between each potential driving factor and FVC in Xiong'an New Area. Finally, considering the policy context and local environmental characteristics, the study provided a reasonable explanation for the change in characteristics of FVC in the Xiong'an New Area from 2005 to 2019.

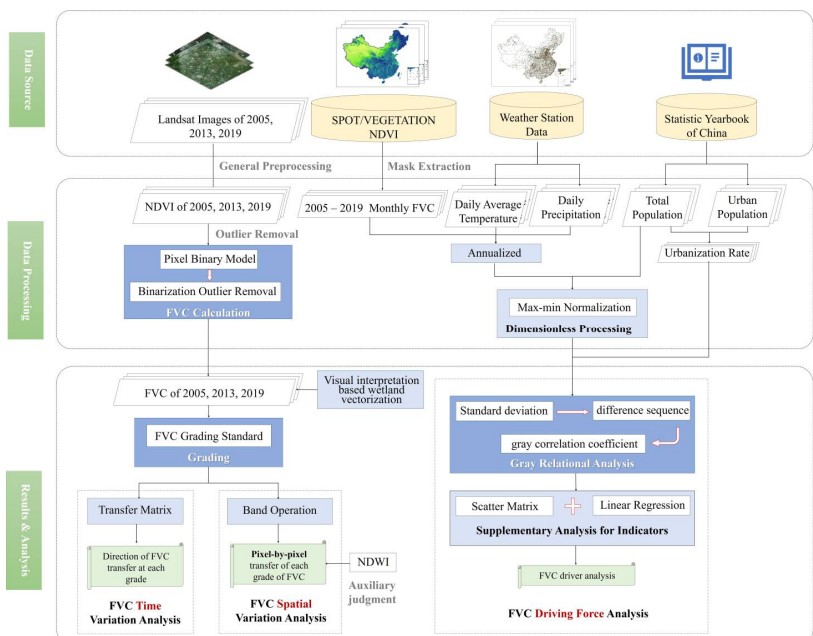

**Figure 2.** The technical route.

### 2.3.1. Processing of Remote Sensing Data

In this study, remote sensing images with less than 5% cloud cover were downloaded for the study area. Landsat 7 images containing bands underwent stripping restoration. Radiometric calibration and atmospheric correction were performed to address differences caused by varying shooting conditions of the remote sensing images during different phases. ENVI (The Environment for Visualizing Images) is a complete, mature, and widely used remote sensing image processing platform, on which we would like to preform radiometric calibration and atmospheric correction process. ENVI 5.3. is the version that we used in this study. At present, atmospheric correction models and methods can be divided into relative correction method based on image features, linear regression method based on ground, atmospheric radiative transfer model, and other composite models [40]. The most famous are FLAASH and 6S, which are both atmospheric radiative transfer models. FLAASH supports a wide range of sensors and adopts the MODTRAN4+ radiation transmission model, which has a high level of accuracy. There is also evidence which shows that the reflectance of local objects is different and the characteristics are obvious in its results, which can better restore the real situation of the surface, and the NDVI response is strong compared to 6S [41]. According to this, FLAASH is suitable for our research on vegetation.

- Radiation calibration

Radiometric calibration was conducted using the radiometric correction tool in ENVI 5.3. This step provided the necessary basis for atmospheric correction. Parameters under the FLAASH method of atmospheric correction were selected and configured.

- Atmospheric correction

For atmospheric correction, the FLAASH method was applied to the radiometrically calibrated images. Various parameters were set, including ground elevation parameters. The values of the digital elevation model (DEM) data within the image range were counted, and the average value was used as the image ground elevation for calculation. Xiong'an, the study area, is located at approximately 39° north latitude, and the analysis was conducted in summer around August. The atmospheric model selected was mid-latitude summer. Based on preliminary visual interpretation of each image, which indicated a dominance of vegetation or cultivated land and a small proportion of urban and construction land, the rustic aerosol model was chosen. The Kaufman aerosol retrieval parameters

were determined as the land surface retrieval standard from 600 nm to 2100 nm in the multispectral setting.

- Mosaic and Clipping

After atmospheric correction, image mosaic and clipping were performed. Only one Landsat image was required to cover the entire study area, resulting in three images for the three study phases. The remote sensing image of the research area was obtained by clipping using authoritative boundary data of Xiong'an provided by the experiment.

### 2.3.2. Dimidiate Pixel Model

The vegetation index serves as the foundation for calculating vegetation coverage. Numerous scholars have conducted research and developed various vegetation index models. Several widely used vegetation index models are available, including RVI, DVI, PVI, NDVI, and DDVI, among others [42]. Among these models, the normalized vegetation index is the most commonly employed (Equation (1)).

$$NDVI = (NIR - R)/(NIR + R) \tag{1}$$

where $NIR$ is the near-infrared band, $R$ is the visible infrared band, and $NDVI$ represents the normalized vegetation index.

Currently, the widely used and applicable estimation method for calculating vegetation coverage is the improved dimidiate pixel model based on NDVI, as proposed by Li [43]. This model allows the pixel NDVI value to be expressed using partial information on vegetation and partial information on bare soil without vegetation coverage, as shown in Equation (2). This model can mitigate the influence of soil background, vegetation type, and the atmosphere, making it suitable for calculating FVC in this study.

$$FVC = \frac{NDVI - NDVI_{soil}}{NDVI_{veg} - NDVI_{soil}} \tag{2}$$

where, $NDVI_{veg}$ represents the partial information of vegetation, and $NDVI_{soil}$ represents the partial information of bare soil without vegetation cover.

The dimidiate pixel model provides vegetation information from the bare land surface, which theoretically should be close to zero and relatively stable. However, due to various real-world conditions, the actual variation range is observed to be between −0.1 and 0.2 [44]. Similarly, high-coverage vegetation information is influenced by factors such as vegetation types and climatic conditions, leading to temporal and spatial changes.

Considering that the study area, Xiong'an, includes the Baiyangdian ecological area, and the analysis is conducted during the summer when vegetation growth is significant, it is determined that NDVI values below 5% represent soil cover. In comparison, values above 95% indicate full vegetation cover. Therefore, NDVI$_{max}$ and NDVI$_{min}$ are chosen as the 95% and 5% frequencies, respectively, in the cumulative probability distribution of NDVI. These values are then substituted into the pixel bisection model to estimate vegetation coverage.

NDVI was calculated using the ENVI software version 5.3 for the three periods of pre-processed remote sensing images. In the case of Landsat 8, which had different band numbers in 2013 and 2019, compared to other Landsat data, the respective red and near-infrared band numbers of 4 and 5 were used in the NDVI formula.

Statistics of NDVI values for each period revealed the presence of outliers outside the typical range. A band operation method was applied to remove outliers and rescale the pixel values to the range of [−1, 1]. Next, the cumulative probability distribution table was used to identify the values close to 5% and 95%, which were considered as the minimum and maximum values of the vegetation index for the respective years. These values were then substituted into the pixel bisection model to calculate vegetation coverage for the three image periods (Table 2).

**Table 2.** Frequency distribution and pixel value of NDVI$_{soil}$ and NDVI$_{veg}$ in each year.

| Year | NDVI$_{soil}$ | | NDVI$_{veg}$ | |
|---|---|---|---|---|
| | Frequency Distribution (%) | Value | Frequency Distribution (%) | Value |
| 2005 | 5.02 | 0.32 | 94.74 | 0.87 |
| 2013 | 5.02 | 0.16 | 95.30 | 0.80 |
| 2019 | 4.86 | 0.15 | 95.41 | 0.91 |

The theoretical value of FVC should fall within the range of [0, 1]. However, due to spectral information from non-vegetation objects such as water bodies, construction land, rocky mountains, and areas with extremely high vegetation coverage in the images, the calculated FVC values may still exceed this interval. To address this issue, the band operation function in ENVI 5.3 was utilized to process the results further using a binary method. This involved assigning values above and below the desired interval to the maximum and minimum values to refine the FVC results.

### 2.3.3. FVC Grading Standard

In order to analyze and compare the changes in FVC during each period in a scientific manner, appropriate criteria were chosen to classify the vegetation coverage results into different grades. Wang et al., based on China's national Soil Erosion Classification and Grading Standard (SL190-2007), as well as considering the actual situation and expert knowledge, established the classification standards for different grades of vegetation coverage in relation to land types. This standard, referred to as the FVC grade classification standard (Table 3), is solely suitable for regions with relatively dry climates [42]. In this study, the vegetation cover classification principle was adopted to classify the FVC results in 2005, 2013, and 2019 in Xiong'an.

**Table 3.** FVC grading standard.

| FVC (%) | Grade | Description of Vegetation Growth and Type |
|---|---|---|
| 0–15 | Bare land | No vegetation, mainly settlements, water areas, unused land, and traffic land |
| 15–30 | Low | Low-yielding grassland, sparse forest land, etc. |
| 30–45 | Mid–low | Middle grassland, woodland, and farmland |
| 45–60 | Medium | High yield grassland, woodland, garden, and farmland, etc. |
| 60–75 | Mid–high | Good grassland, shrub land, woodland, and good arable land, etc. |
| >75 | High | Dense shrub land, dense forest, etc. |

### 2.3.4. Normalized Difference Water Index

Normalized difference water index (NDWI) is a widely used method to highlight water information in the image (Equation (3)) [45]. The range of NDWI should be [−1, 1], which is the same as that for NDVI. Generally, the discriminant threshold of water body is 0.2, namely, the area could be regarded as water if its NDWI value is beyond 0.2. Here, we used NDWI to assist in judging FVC patterns and vectorizing a boundary of the Baiyangdian wetland.

$$NDWI = (Green - NIR)/(Green + NIR) \tag{3}$$

where Green is the green band, NIR is the near-infrared band, and NDWI represents the normalized water body index.

### 2.3.5. The Transition Matrix

The Transition Matrix reflects the conversion relationships between different land use types within a specific area over a certain period (Equation (4)). It represents the area of land that changes from one land use type to another within the defined timeframe,

measured in square kilometers [46]. We constructed transition matrices for different FVC grades in the years 2005, 2013, and 2019.

$$S_{ij} = \begin{bmatrix} S_{11} & \dots & S_{1n} \\ \vdots & & \vdots \\ S_{n1} & \dots & S_{nn} \end{bmatrix} \tag{4}$$

where, $S_{ij}$ represents the area in square kilometers of land use type $i$ that undergoes a transition to land use type $j$ during a specific period. $n$ denotes the total number of land use types considered in the analysis.

In our study, we utilized ArcMap 10.6 and the Tabulate Areas tool to calculate the transition matrix for vegetation cover levels of the 2005, 2013, and 2019 FVC raster images, which were categorized into different vegetation cover levels.

2.3.6. Grey Relational Analysis

The grey correlation degree analysis method is based on grey system theory. Its primary objective is to analyze the degree of difference in geometric shapes between curves and quantify and compare the correlation coefficients. This method is particularly suitable for analyzing time series or dynamic historical data with variables exhibiting time trends [47].

Several potential factors influence FVC, including precipitation, temperature, human activities, and other aspects. These factors collectively form a grey system characterized by incompleteness and uncertainty. Moreover, the grey correlation degree analysis method is applicable even when dealing with small sample sizes, defined as sample sizes less than 30. Considering these factors, this method is highly suitable for driving force analysis in our study. To conduct the analysis, we annualized the SPOT/VEGETATION NDVI data, which consisted of monthly FVC values from 2005 to 2019 calculated using the same method applied to the three Landsat images. We also included daily average temperature, daily precipitation, total population at the end of each year, and urbanization rate in the dataset. Subsequently, we performed grey relational analysis using the following main steps:

- Determine the reference and comparison sequences

Firstly, we selected the data sequences to be used as a reference and for comparison in correlation analysis. The reference sequence in this study is the FVC. The data sequences to be analyzed include the urbanization rate, the total population of the region at the end of each year, the cumulative precipitation from 20:00 to 20:00 the next day, and the temperature.

- Nondimensionalize

In the second step, we must compare the data by applying dimensionless processing before conducting the correlation analysis. Since the original data dimensions and scales of different variable sequences vary, dimensionless processing is necessary. There are four widely used methods for dimensionless processing, namely the mean method, extremization method, standardization method, and standard deviation method. In this study, the objective is to compare the similarity and fit between the change trends of each variable and FVC in order to analyze their main driving force. Therefore, we have selected the standardization method (Equation (5)) for dimensionless processing. This method helps eliminate the influence of dimensions and magnitudes and the differences in variation degrees among variables. It ensures that each variable carries equal importance in the subsequent analysis after transformation [48].

$$x' = \frac{x - \overline{x}}{s} \tag{5}$$

where $x$ is each variable, $\overline{x}$ is its mean value, $s$ is the standard deviation of the variable, and $x'$ is the sample obtained after standardization.

- Calculate the grey correlation degree of sequences

In this step, we calculate the grey correlation degree between the reference sequence and the comparison sequence. The change trend of each sequence is represented by the spacing between corresponding points. If the spacing between two points is small, indicating a small difference, it indicates a high consistency in the change trend of the two sequences. On the other hand, if the spacing between two points is large, indicating a large difference, it indicates a low consistency in the change trend of the two sequences (Equation (6)) [49].

$$\Delta_{0i}(k) = |x_0(k) - x_i(k)|$$
$$(k = 1, \ldots, m; i = 1, \ldots, n)$$
(6)

where $x_0$ is the reference sequence, $x_i$ is the comparison sequence, $\Delta_{0i}(k)$ is the distance between samples of the reference sequence and the comparison sequence, and $m$ and $n$ are the sample size and the number of evaluated sequences, respectively.

In theory, the correlation coefficient of each sequence can be calculated by taking the average value of the difference sequence. However, in practical situations, there are often significant differences in the magnitude of each sequence. In such cases, it is necessary to introduce the minimum value (represented by Equation (7)) and maximum value (represented by Equation (8)) from the difference sequence. These values are then substituted into the improved formula (represented by Equation (9)) for calculating the correlation coefficient.

$$x_{min} = \min_{i=1}^{n} \min_{k=1}^{m} |x_0(k) - x_i(k)|$$
(7)

$$x_{max} = \max_{i=1}^{n} \max_{k=1}^{m} |x_0(k) - x_i(k)|$$
(8)

$$\zeta_i(k) = \frac{\min_i \min_k |x_0(k) - x_i(k)| + \rho \cdot \max_i \max_k |x_0(k) - x_i(k)|}{|x_0(k) - x_i(k)| + \rho \cdot \max_i \max_k |x_0(k) - x_i(k)|}$$
(9)

where, in Equations (7) and (8), $x_{min}$ is the minimum value and $x_{max}$ is the maximum value. In Equation (9), $\zeta_i(k)$ is the correlation coefficient of each variable after re-normalization, $\rho$ is the discrimination coefficient, and the value interval is (0, 1).

The value of $\rho$ plays a role in controlling the impact of the maximum value on the transformation result. When $\rho$ is small, it increases the difference in correlation coefficient between each point, thereby enhancing the discriminative ability. Typically, the default value for $\rho$ is set to 0.5 [50].

- Calculate the association order

The average correlation degree of each comparison sequence is calculated, and the correlation order between each comparison sequence and the reference sequence is determined. This result is denoted as $r_{oi}$ (Equation (10)) and referred to as the grey correlation degree between the sequences $r_o$ and $r_i$. In this study, the correlation orders of the urbanization rate, regional total population at the end of the year, cumulative precipitation from 20:00 to 20:00 the next day, and temperature are denoted as $r_{o1}$, $r_{o2}$, $r_{o3}$, and $r_{o4}$, respectively.

$$r_{oi} = \frac{1}{m} \sum_{k=1}^{m} \zeta_i(k)$$
(10)

### 2.3.7. Linear Regression

In driving force analysis, linear regression has been widely used as a practical and common method [51–53]. To further investigate the positive or negative effects of each factor on FVC, we conducted a supplementary analysis using the linear regression method. Since the calculation of the grey correlation degree involved the standard deviation sequence of each factor, and the results of simple linear regression are not affected by data standardization,

we directly performed linear regression analysis on the standardized deviation sequence of each factor. By examining the coefficient $\beta_1$ of the independent variable in the regression equation (Equation (11)), we can determine the positive or negative effects of the driving forces on FVC.

$$Y = \beta_0 + \beta_1 x + \varepsilon \tag{11}$$

where $\beta_0$ is the constant coefficient, which represents the intercept or the value of $Y$ when $x$ is zero. $\beta_1$ is the coefficient of the independent variable, which measures the change in $Y$ for a unit change in $x$. $\varepsilon$ represents the uncertainty or error in the relationship between $Y$ and $x$, capturing the part of $Y$ that is not explained by the independent variable.

In our linear regression analysis, the dependent variable ($Y$) represents FVC, which is the variable we want to predict or explain. The independent variable ($x$) represents each potential driving force, such as population, urbanization rate, average annual temperature, and precipitation.

## 3. Results

### 3.1. General FVC Characteristics of Xiong'an from 2005 to 2019

#### 3.1.1. Spatial Distribution

The three-phase FVC results of the three counties in the Xiong'an New Area exhibit distinct spatial distribution differences, with noticeable changes in local FVC from 2005 to 2019 (Figure 3). The expansion of large-scale urban areas has decreased vegetation cover in the surrounding areas, such as the western part of Rongcheng County, the western part of Xiongxian County, and the northern part of Anxin County. However, the degradation of vegetation coverage around smaller towns and villages occurred in patchy patterns, primarily on the periphery of the three counties in Xiong'an, particularly in villages far from the main urban centers. In 2019, the FVC distribution indicated that low FVC concentrations were predominantly located in the southwest of the starting area, which is situated in the northwest region of the overall area, based on the functional area division of Xiong'an.

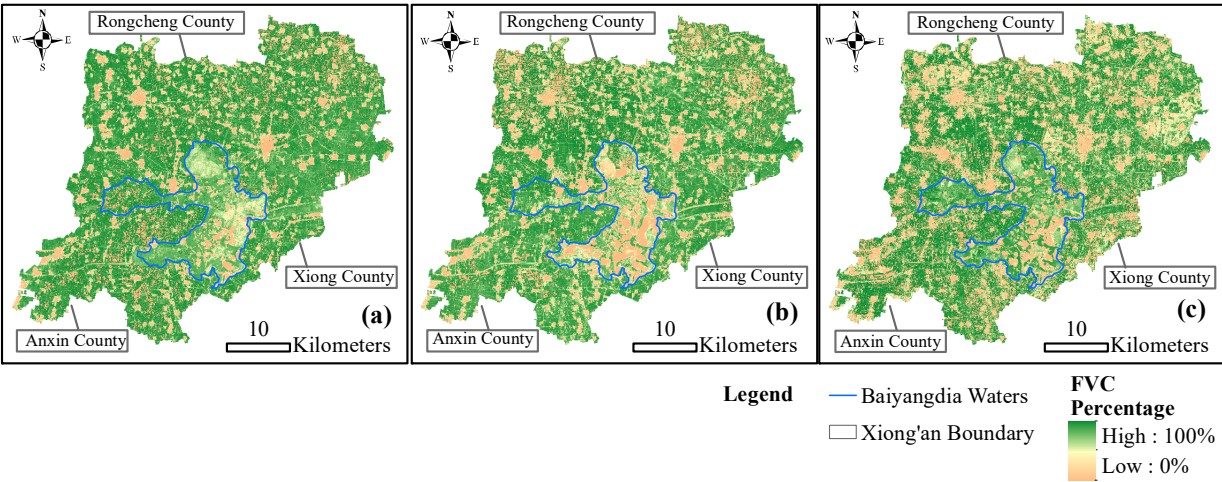

**Figure 3.** Xiong'an FVC in the year of (**a**) 2005, (**b**) 2013, (**c**) 2019.

The land cover analysis in Xiong'an New Area reveals that cropland and grassland are the dominant vegetation cover types (Figure 4). During the study period, the impervious surfaces increased from 9.74% to 11.88%, showing a significant growth rate. Water bodies initially increased but later showed a declining trend. Wetlands exhibited continuous growth. Grassland and cropland experienced slight decreases, but the magnitude was relatively small. Forest and grassland remained relatively stable, although their individual proportions might be very low, as collectively they did not exceed 0.15% of the area. These results to some extent confirm signs of urban expansion and changes in wetland areas within the region.

**Land use/cover**

**Figure 4.** The frequency of different land use/cover during the study period.

In order to analyze the vegetation changes in Baiyangdian wetland, a significant local ecological area, we used visual interpretation based on the original images of each year, as well as NDVI, FVC, and NDWI data. Referring to the boundary of the Baiyangdian ecological area established in previous studies, we delineated the water concentration area of Xiong'an Baiyangdian Wetland and determined its boundary range [54]. The FVC of Baiyangdian wetland exhibited a general pattern of initial decline and a subsequent slight improvement over the three years, with the highest value in 2005 and the lowest value in 2013. In 2019, there were low FVC areas with fragmented distribution. By analyzing the normalized water index (NDWI), we found that the areas with low FVC values in Baiyangdian Lake in 2013 generally had NDWI values at approximately 0.5 and greater than 0.3, indicating that water was the predominant feature.

3.1.2. The Change Pattern of FVC Grades

According to the statistics of FVC area in the study area for the three stages, it is evident that the three counties predominantly consist of bare land, with an average proportion of 54.06%. High-coverage vegetation follows with an average standing ratio of 33.08%. The remaining grades, from low to high coverage, account for 2.67%, 2.54%, 2.97%, and 4.69%, respectively. Vegetation with medium coverage and below experienced continuous growth over the 15 years, with average growth rates of 3.08%, 39.86%, 30.05%, and 15.76% respectively. Vegetation with mid–high coverage initially increased and then decreased, while high-coverage vegetation showed a degradation trend with an average annual degradation rate of 10.34% (Figure 5a). It can be observed that bare land is the most stable vegetation coverage class with the largest proportion in the local area, followed by high-coverage vegetation. From 2005 to 2019, vegetation grades from low to medium increased steadily, with the proportion of increase decreasing with higher grade improvement, while vegetation with high coverage exhibited a recent degradation trend. Each coverage grade's FVC map of the three years are also given (Figure 5b–d), which reflect similar characteristics with the original FVC results (Figure 4).

There are certain discrepancies between the FVC histogram and land cover statistics (Figures 4 and 5a), particularly in bare land. This could be attributed to differences in the interpretation of grassland by specific methods. Distinguishing between grassland and cropland can also pose challenges. Furthermore, the relatively low proportion of forests and the overall higher FVC are not contradictory, as crops can also contribute to a higher FVC. In fact, the actual situation of various land surface cover types requires more precise and verifiable analysis.

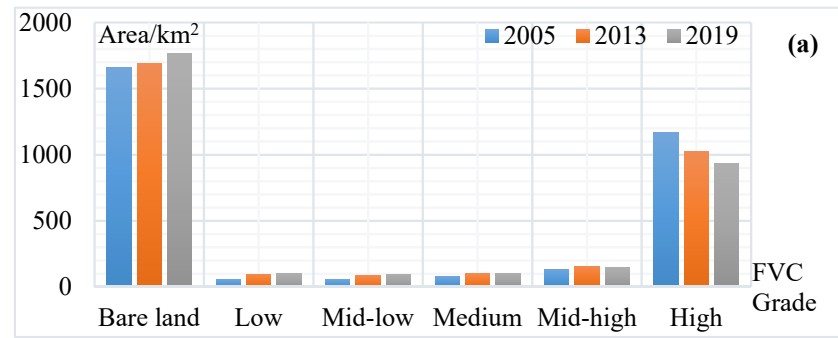

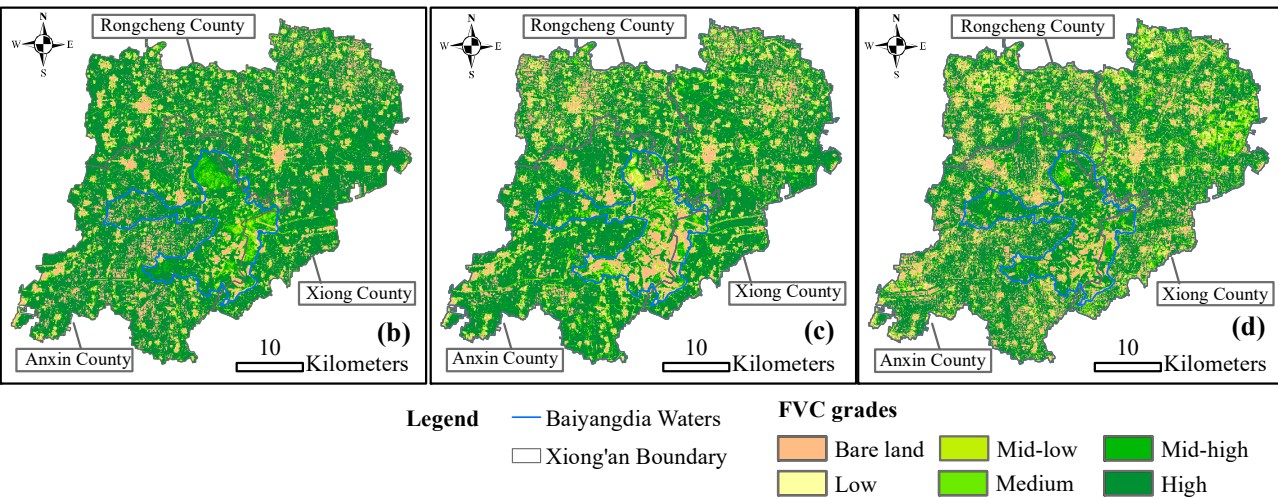

**Figure 5.** Each FVC grade's change patterns in Xiong'an New Area from 2005 to 2019: (**a**) Statistical histogram of FVC area; (**b**) FVC spatial patterns in 2005; (**c**) FVC spatial patterns in 2013; (**d**) FVC spatial patterns in 2019.

*3.2. Pixel-by-Pixel Spatial Variation of FVC*

3.2.1. The Pixel-by-Pixel Transfer Characteristics of FVC from 2005 to 2013

The change in FVC per pixel was calculated using the band operation subtraction method, and the results were divided into seven grades using the equal interval method. From 2005 to 2013, the FVC in most Xiongxian, Anxin, and Rongcheng counties remained relatively stable (Figure 6a). However, this needed to be confirmed with further quantitative analysis, which we conducted in the next step. Vegetation degradation was primarily concentrated in the northwest of Rongcheng County, the wet areas of Baiyangdian Lake, and the east of Xiongxian County. These degradation patterns occurred in patchy formations and water areas, respectively. Additionally, there was a noticeable phenomenon of FVC degradation along roads during this stage. The degradation extended widely along the roads but did not diffuse beyond that (Figure 6c).

On the other hand, vegetation improvement predominantly occurred in the western to southern areas of Baiyangdian Lake and in the northern part of Anxin County. These improvements were mainly observed in fragmented patches (Figure 6d). In the Baiyangdian ecological area, the FVC did not exhibit a significant increase from 2005 to 2013, for there was mainly balanced area. The decline in FVC in this area was primarily attributed to the increase in water bodies, as indicated by NDWI (Figure 6b), which is consistent with existing relevant research [7].

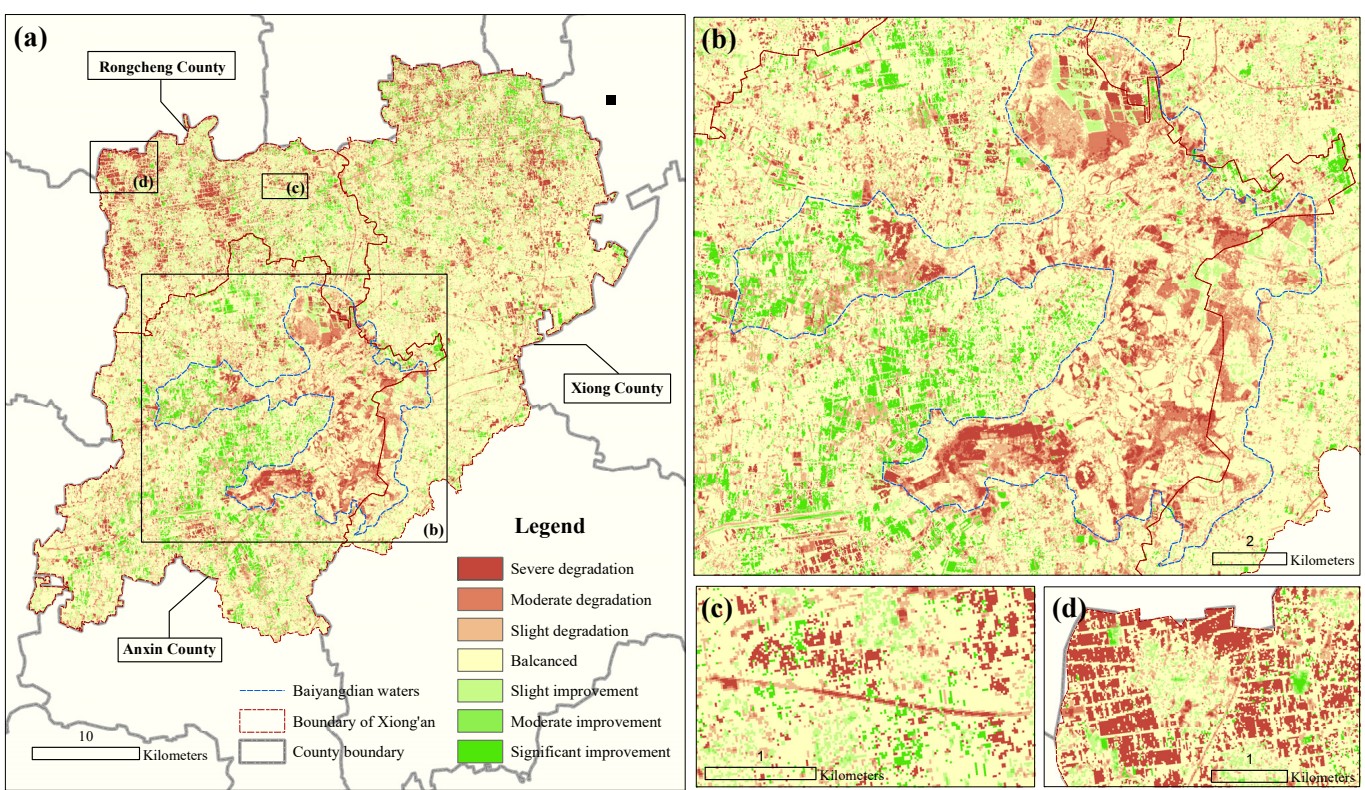

**Figure 6.** Pixel-by-pixel spatial transfer of FVC in Xiong'an from 2005 to 2013: (**a**) the whole study area; (**b**) the Baiyangdian waters; (**c**) degradation along the road; (**d**) patchy degradation.

3.2.2. The Pixel-by-Pixel Transfer Characteristics of FVC from 2013 to 2019

From 2013 to 2019, the FVC in Xiong'an exhibited a more fragmented degradation trend, characterized by numerous uniformly distributed degraded patches in each region. However, these patches' size was smaller than in the previous stage (Figure 7a).

The linear degradation of FVC caused by roads decreased in comparison to the previous stage. Only a few individual lines were observed in the west and south, while a diffuse degradation area was present in the northeast. Upon examining the relevant map data, it was determined that the line corresponded to the "Beijing-Xiong intercity" railway line, which connected the R1 line of Beijing Daxing Airport Express in the center of Xiongxian County in Xiong'an (Figure 7c).

The improvement of FVC during this stage primarily occurred in the Baiyangdian wetland area and the northern part of the three counties. The improvement in the northern region still occurred in patchy patterns (Figure 7d). The FVC changes in the Baiyangdian wetland area mainly exhibited significant improvements, with little to no degradation. This area served as the main location for FVC improvement during this stage and displayed the characteristic of improvement around water (Figure 7b). However, for wetland ecosystems, an increase in surface vegetation coverage often signifies a decrease in water content, and the original aquatic vegetation is replaced by herbaceous shrubs or even trees with higher coverage. Therefore, the recent increase in FVC in the Baiyangdian Lake may be related to the degradation of the wetland environment.

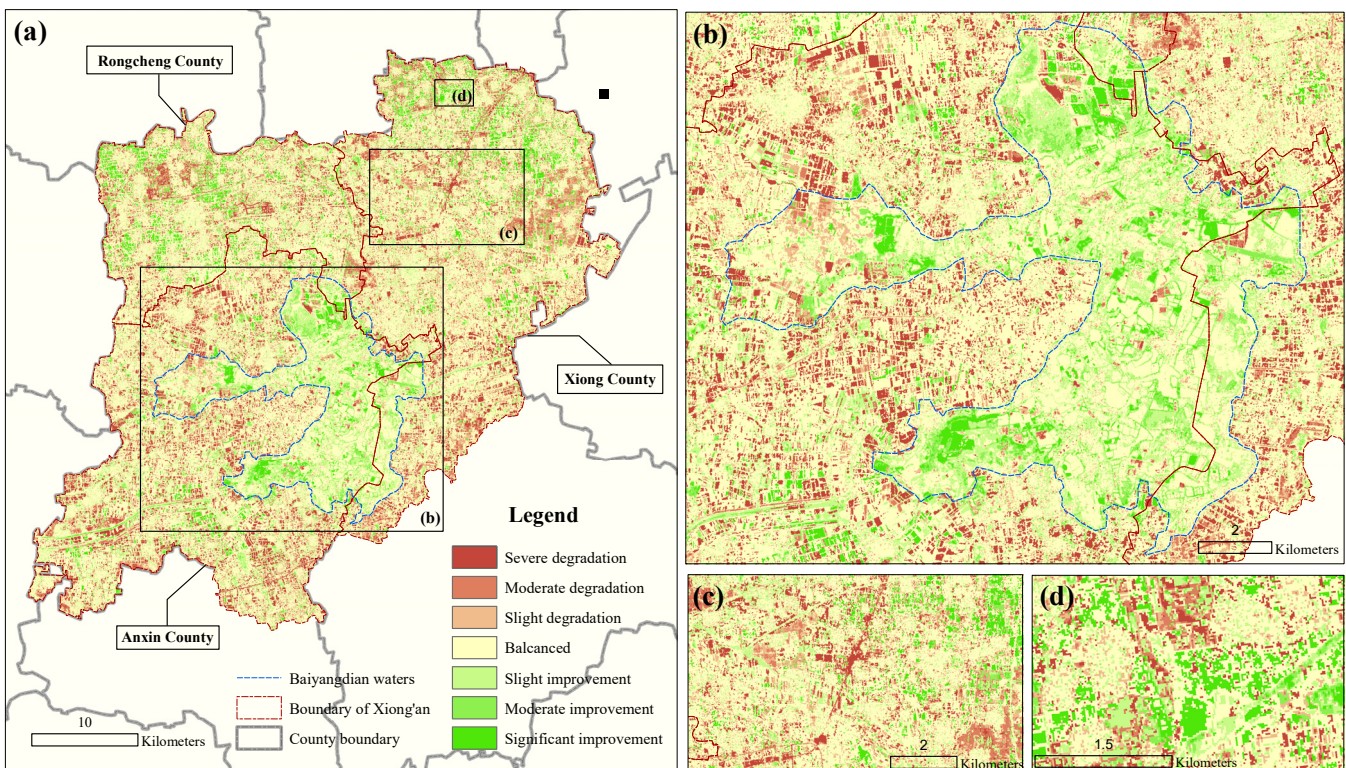

**Figure 7.** Pixel-by-pixel spatial transfer of FVC in Xiong'an from 2013 to 2019: (**a**) the whole study area; (**b**) the Baiyangdian waters; (**c**) part of the line R1 of Beijing Daxing airport express of "Beijing-Xiong intercity" railway line in Xiong'an; (**d**) improved area.

3.2.3. Overall Spatial Transfer Characteristics

Overall, the changes in vegetation cover in the three counties of Xiong'an from 2005 to 2019 exhibited a mixture of degradation and improvement, with a relatively more pronounced degradation phenomenon (Figure 8a,c). The areas of improvement were fewer and mainly longitudinally distributed along the northeast to southwest direction of Xiong'an. On the other hand, the degradation areas were uniformly distributed in patchy formations across the entire region (Figure 8d).

The Baiyangdian wetland had the largest FVC balance area, and its vegetation was relatively more stable. However, the main change observed in this area was the decline in FVC, and the degradation patterns appeared as patches or around water bodies (Figure 8b). It could be some ecological protection efforts in the Baiyangdian wetland have achieved certain results, including a more comprehensive institutional mechanism for ecosystem conservation, restoration, and management. However, further efforts are needed to ensure the continuous restoration and stability of the ecological environment in the area.

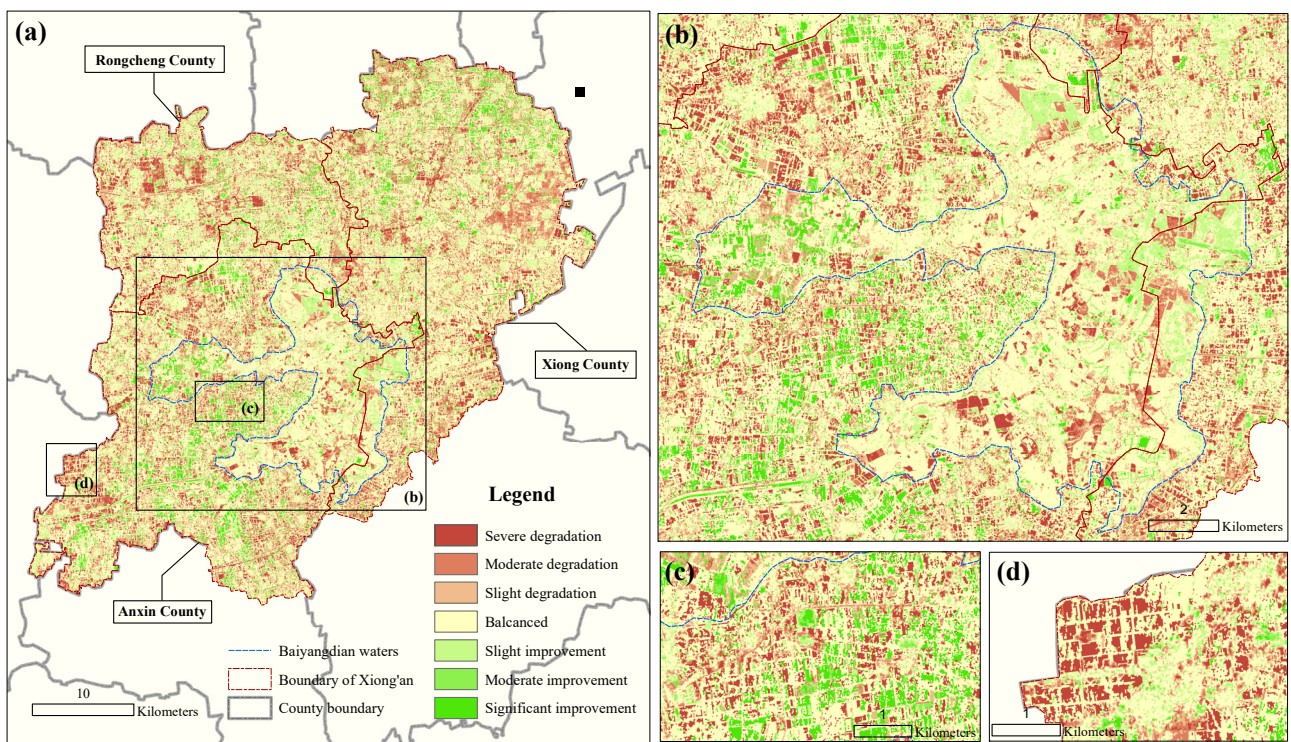

**Figure 8.** Pixel-by-pixel spatial transfer of FVC in Xiong'an from 2005 to 2019: (**a**) the whole study area; (**b**) the Baiyangdian Waters; (**c**) the intersection area of degradation and improvement; (**d**) patchy degeneration.

*3.3. The Transformation Direction of FVC at Different Grades*

3.3.1. FVC Transfer Direction at Each Grade from 2005 to 2013

From 2005 to 2013, the transfer pattern remained consistent with the overall trend. The bidirectional conversion ratio of each grade gradually increased or decreased with the corresponding increase or decrease in the grade, but there was a sharp increase in the conversion to bare land. Vegetation with lower coverage was more likely to be transformed into bare land than vegetation with higher coverage (Table 4). Except for the highest grade, each grade's overall positive conversion rates were 56.81%, 35.39%, 32.69%, 32.04%, and 30.86%, respectively. Except for bare land, the negative conversion rates for each grade were 32.81%, 43.89%, 48.19%, 46.86%, and 26.58%, respectively. On average, the positive and negative conversion rates were 37.56% and 39.67%, respectively.

**Table 4.** 2005–2013 FVC transition matrix of different grades in Xiong'an. (Area/km$^2$, Rate/%).

| 2013 | 2005 | | | | | | | | | | | |
|---|---|---|---|---|---|---|---|---|---|---|---|---|
| | **Bare Land** | | **Low** | | **Mid–Low** | | **Medium** | | **Mid–High** | | **High** | |
| | **Area** | **Rate** | **Area** | **Rate** | **Area** | **Rate** | **Area** | **Rate** | **Area** | **Rate** | **Area** | **Rate** |
| Bare land | 124.87 | 43.19 | 18.07 | 32.81 | 13.14 | 22.73 | 14.88 | 19.03 | 20.35 | 15.11 | 124.35 | 10.66 |
| Low | 29.67 | 10.26 | 17.51 | 31.80 | 12.23 | 21.17 | 9.58 | 12.26 | 9.58 | 7.11 | 17.32 | 1.49 |
| Mid–low | 10.02 | 3.46 | 11.31 | 20.53 | 13.53 | 23.41 | 13.21 | 16.90 | 13.31 | 9.88 | 25.57 | 2.19 |
| Medium | 5.51 | 1.91 | 4.40 | 7.99 | 9.66 | 16.71 | 15.46 | 19.78 | 19.88 | 14.76 | 43.66 | 3.74 |
| Mid–high | 7.71 | 2.67 | 2.02 | 3.67 | 5.48 | 9.48 | 14.36 | 18.37 | 30.00 | 22.27 | 99.11 | 8.50 |
| High | 111.36 | 38.51 | 1.76 | 3.20 | 3.76 | 6.50 | 10.69 | 13.67 | 41.57 | 30.86 | 856.39 | 73.42 |

It is evident that during this stage, the local vegetation underwent significant improvement and degradation, resulting in an unstable FVC. Bare land displayed a noticeable upward trend, with relatively minimal degradation observed in the highest grade. Upon examining the diagonal data, the stability rates for all grades were as follows: 43.19%,

31.80%, 23.41%, 19.78%, 22.27%, and 73.42%, respectively. The highest coverage grade remained the most stable still, followed by bare ground. Compared to the overall situation, there was an improvement in all stability rates, indicating that the FVC during the first stage was relatively stable.

3.3.2. FVC Transfer Direction at Each Grade from 2013 to 2019

From 2013 to 2019, there was a significant decrease in the negative conversion rate and a substantial increase in the positive conversion rate for each grade compared to the previous stage. On average, the positive conversion rate was 51.42%, while the negative conversion rate was 26.66%. The improvement rate was twice as high as the negative degradation rate. Additionally, there was a notable improvement in the direct conversion ratio to the highest grade for all grades (Table 5). During this stage, the proportion of high-coverage vegetation converted to bare land increased from 10.66% in the previous stage to 17.52%. This resulted in a degraded area of 179.63 km², representing this stage's largest area transfer.

**Table 5.** 2013–2019 FVC transition matrix of different grades in Xiong'an. (Area/km², Rate/%).

| 2019 | 2013 | | | | | | | | | | | |
| | Bare Land | | Low | | Mid–Low | | Medium | | Mid–High | | High | |
| | Area | Rate | Area | Rate | Area | Rate | Area | Rate | Area | Rate | Area | Rate |
| --- | --- | --- | --- | --- | --- | --- | --- | --- | --- | --- | --- | --- |
| Bare land | 159.43 | 50.51 | 26.49 | 27.63 | 9.79 | 11.26 | 6.91 | 7.01 | 10.58 | 6.66 | 179.63 | 17.52 |
| Low | 28.44 | 9.01 | 29.58 | 30.85 | 13.99 | 16.09 | 5.78 | 5.87 | 3.73 | 2.35 | 19.75 | 1.93 |
| Mid–low | 14.07 | 4.46 | 16.43 | 17.13 | 19.81 | 22.79 | 11.41 | 11.57 | 6.92 | 4.36 | 26.71 | 2.60 |
| Medium | 12.78 | 4.05 | 7.58 | 7.90 | 15.59 | 17.94 | 17.88 | 18.13 | 13.77 | 8.68 | 36.29 | 3.54 |
| Mid–high | 15.12 | 4.79 | 6.75 | 7.04 | 12.02 | 13.82 | 22.02 | 22.34 | 30.37 | 19.14 | 64.29 | 6.27 |
| High | 85.81 | 27.19 | 9.07 | 9.46 | 15.74 | 18.11 | 34.58 | 35.08 | 93.31 | 58.81 | 698.85 | 68.15 |

The diagonal data indicates that the stability rates for all grades in this stage were as follows: 50.51%, 30.85%, 22.79%, 18.13%, 19.14%, and 68.15%, respectively. Compared to the previous stage, bare land tended towards greater stability, while the degradation of the highest grade slightly intensified.

3.3.3. Overall Temporal Transfer Characteristics

Over the past 15 years, approximately 33.04% of the bare land in Xiong'an has transformed into high-coverage vegetation. The positive conversion ratio of low-coverage vegetation decreased as the grade increased. Similarly, the positive and negative conversion ratio of medium-coverage vegetation exhibited a synchronous relationship with the upgrading or downgrading the grade. Notably, 43.86% of the areas originally classified as mid–high coverage vegetation further improved their FVC and transitioned into high-coverage vegetation areas. This transition constituted the main source of the increase in high coverage (Table 6).

**Table 6.** 2005–2019 FVC transition matrix of different grades in Xiong'an. (Area/km², Rate/%).

| 2019 | 2005 | | | | | | | | | | | |
| | Bare Land | | Low | | Mid–Low | | Medium | | Mid–High | | High | |
| | Area | Rate | Area | Rate | Area | Rate | Area | Rate | Area | Rate | Area | Rate |
| --- | --- | --- | --- | --- | --- | --- | --- | --- | --- | --- | --- | --- |
| Bare land | 129.13 | 44.66 | 19.43 | 35.29 | 14.20 | 24.57 | 13.53 | 17.31 | 17.08 | 12.68 | 199.46 | 17.10 |
| Low | 30.11 | 10.41 | 14.19 | 25.77 | 10.35 | 17.91 | 8.00 | 10.23 | 7.04 | 5.23 | 31.57 | 2.71 |
| Mid–low | 14.06 | 4.86 | 9.80 | 17.81 | 10.58 | 18.30 | 9.94 | 12.71 | 9.93 | 7.37 | 41.03 | 3.52 |
| Medium | 9.30 | 3.22 | 5.22 | 9.47 | 8.53 | 14.76 | 12.21 | 15.62 | 14.95 | 11.10 | 53.68 | 4.60 |
| Mid–high | 11.00 | 3.80 | 2.90 | 5.27 | 6.80 | 11.76 | 15.27 | 19.53 | 26.61 | 19.76 | 87.99 | 7.54 |
| High | 95.54 | 33.04 | 3.52 | 6.39 | 7.34 | 12.70 | 19.23 | 24.61 | 59.07 | 43.86 | 752.65 | 64.53 |

Regarding negative transformation for each grade, the highest proportion of transformation occurred in the bare land category. The conversion rates which are from mid–low and low grades into bare land were 24.57% and 35.29%, respectively, significantly higher than the rates of 12.68% and 17.10% for mid–high and high-coverage vegetation. This indicates that areas with lower-grade vegetation coverage are more susceptible to degradation. Furthermore, the area of bare land converted from high-coverage vegetation was nearly 200 km$^2$, 1.5 times the stable area of bare land.

Except for the highest grade, each grade's overall positive conversion rates were 55.34%, 38.94%, 39.22%, 44.13%, and 43.86%, respectively. The negative conversion rates for all grades, except for bare soil, fluctuated within the range of 35–40%. This suggests that each grade's positive and negative transfer rates are relatively stable, while vegetation cover exhibits strong fluctuations, and bare land is more likely to undergo a positive transformation.

By examining the diagonal data, the stable rates for each grade are as follows: 44.66%, 25.77%, 18.30%, 15.62%, 19.76%, and 64.53%, respectively. This indicates that the highest grade and bare land are the most stable FVC.

### 3.4. Grey Relational Analysis Results

The grey relational analysis results do not have strict and standardized classification criteria or empirical judgments. Our assessment is mainly based on comparisons to determine the degree of association between different factors and the independent variable. The results of the grey correlation analysis reveal that each factor exhibits a correlation coefficient above 0.5 with FVC. Considering the value range of the grey correlation results, which spans from 0 to 1, it can be inferred that each factor demonstrates a certain degree of correlation with FVC. Notably, annual precipitation exhibits the highest correlation coefficient of 0.760, followed by population, temperature, and urbanization rate, which hover at approximately 0.6 (Table 7). When examining the standardized data series line plots for each factor (Figure 9), it becomes evident that FVC shows the strongest fit with annual precipitation. The four factors were also categorized as natural factors (precipitation and temperature) and human factors (urbanization rate and population). The average correlation coefficient between natural factors and FVC was 0.67, while the average correlation coefficient between human factors and FVC was 0.60.

**Table 7.** Grey relational analysis result.

| Factors * | Relational Order | Relational Value | Ranking |
|:---:|:---:|:---:|:---:|
| AP | $r_{o1}$ | 0.760 | 1 |
| TP | $r_{o3}$ | 0.613 | 2 |
| AMT | $r_{o4}$ | 0.589 | 3 |
| UR | $r_{o2}$ | 0.587 | 4 |

* TP: Total Population; UR: Urbanization Rate; AMT: Annual Mean Temperature; AP: Annual Precipitation.

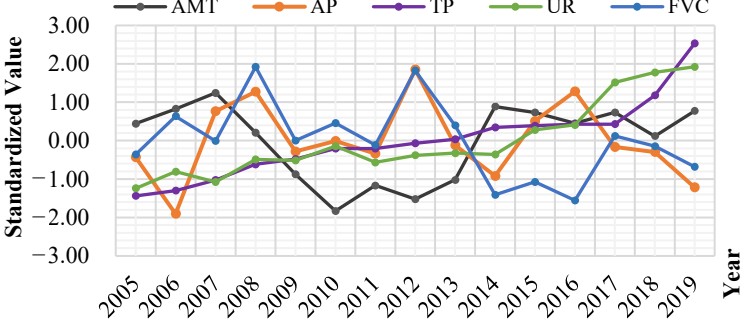

**Figure 9.** Standardized sequences lists of FVC and factors. FVC: Fractional Vegetation Coverage; TP: Total Population; UR: Urbanization Rate; AMT: Annual Mean Temperature; AP: Annual Precipitation.

Due to the nature of the grey relational analysis, which focuses solely on the morphological aspect of factor correlations without considering interaction effects, we can directly determine that in the period from 2005 to 2019, precipitation shows the closest relationship with vegetation in Xiong'an New Area. The variation trend of FVC exhibits better consistency with precipitation compared to temperature, population, and urbanization rate. This implies that an increase in precipitation often indicates favorable vegetation growth. This trend change can be promptly reflected in the same year without lag. Moreover, a more accurate trend change also indicates a more predictive value of precipitation for FVC compared to other factors.

### 3.5. Driving Force Analysis Based on the Linear Regression

The simple univariate linear regression results indicate that population, urbanization rate, and average annual temperature are negatively correlated with FVC, while precipitation exhibits a positive correlation. These findings align with common sense expectations (Figure 10).

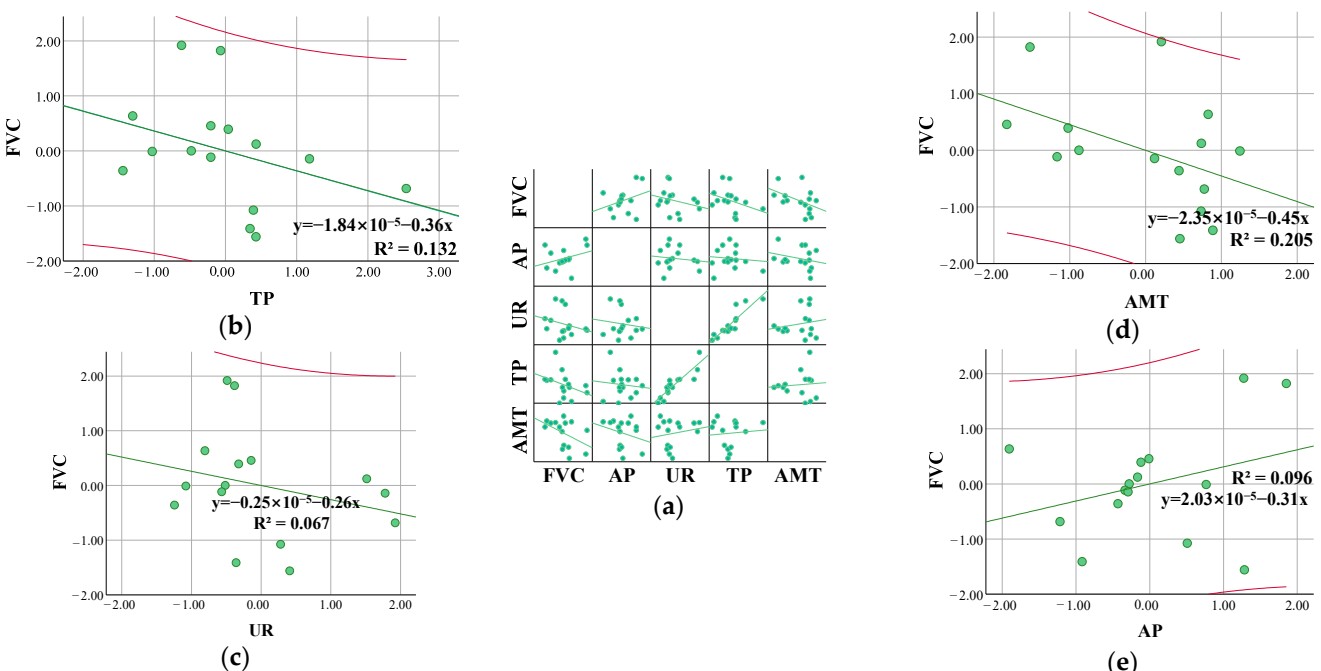

**Figure 10.** Scatter plots and linear regression plots of FVC with its factors: (**a**) Scatters matrix of FVC and TP, UR, AMT, and AP. (**b**) Linear regression of FVC and TP. (**c**) Linear regression of FVC and UR. (**d**) Linear regression of FVC and AMT. (**e**) Linear regression of FVC and AP where, FVC: Fractional Vegetation Coverage; TP: Total Population; UR: Urbanization Rate; AMT: Annual Mean Temperature; AP: Annual Precipitation.

We also found that the most influential factor correlated with FVC is the AMT (Annual Mean Temperature), as it exhibits the highest $R^2$ value of 0.205. The subsequent factors in descending order of importance are TP (Total Population), UR (Urbanization Rate), and AP (Annual Precipitation). Other scholars have also found that the $R^2$ of the linear regression between precipitation, temperature, and vegetation is approximately 0.2 [55]. Moreover, precipitation exhibits better correlation with vegetation compared to temperature, with a higher $R^2$ value [56]. In scenarios with more frequent human activities, the $R^2$ value can reach approximately 0.5. This to some extent confirms the reliability of our work and suggests that the human activities currently have a relatively weak impact on vegetation in Xiong'an.

Therefore, we only employed linear regression analysis to ascertain the positive and negative effects of each driving force on FVC, providing a valuable complement to the results of the grey correlation analysis.

## 4. Discussion

### 4.1. Comparison Validation

Based on the calculated annual Fractional Vegetation Cover (FVC) time series data from SPOT imagery, the trend lines indicate that during the time period, the most significant decreases in FVC occurred in the years 2008–2009 and 2012–2014, with a decline rate of 7.4% and 4.9%, respectively (Figure 11). The overall vegetation cover in the area tends to be relatively high, ranging from 0.74 to 0.82. Over the study period, there are slight fluctuations and a slight decreasing trend observed in the FVC values. The average FVC values in Xiong'an for 2005, 2013, and 2019 were 0.75, 0.74, and 0.71, respectively, indicating a downward trend. The FVC values for the corresponding years, derived from the long-term series vegetation index data, were 0.76, 0.78, and 0.76, respectively. That is to say, the three-year trend in FVC exhibited consistency with the overall and predicted trends. In summary, the resulting data align with the actual observations, reinforcing the credibility of the analysis' findings and conclusions.

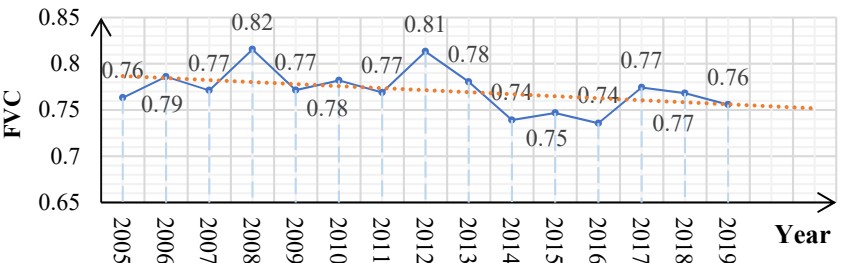

**Figure 11.** FVC based on SPOT/VEGETATION NDVI with a trend line.

### 4.2. Result Discussion

During the period from 2005 to 2019, which largely encompassed the pre-planning phase, the grade structure of FVC in Xiong'an was predominantly composed of bare land, followed by high-coverage vegetation, with a scarcity of natural forests in the area. Therefore, vegetation degradation and improvement primarily occurred in patchy patterns. The variability in agricultural land planting played a significant role in influencing local vegetation cover. Degraded linear area indicated that the construction of transportation infrastructure, industrial development, and other projects impacted vegetation, which was very limited but significant.

Throughout the study period, there were significant fluctuations in FVC. However, in recent years, a trend of increasing positive transformations has emerged overall, indicating an improvement in the ecological environment. Lower-grade FVC areas are more susceptible to degradation and improvement, while high-coverage vegetation exhibits greater stability. The expansion of the high-coverage vegetation area stems from mid–high and bare land. Therefore, local efforts should focus on protecting areas with low vegetation coverage while promoting the enhancement of overall vegetation cover to achieve a balance between stability and ecological optimization. Simultaneously, attention should be given to the stable preservation of existing areas with high vegetation FVC, ensuring that rapid development and the construction of ecological civilization proceed harmoniously.

During the initial stage, the key ecological area of Baiyangdian wetland experienced an increase in water bodies and a decrease in vegetation, while it subsequently had an improvement. Baiyangdian exhibits relatively small fluctuations in FVC within the Xiong'an New Area, indicating a comparatively stable ecological environment. The above results collectively indicate that the development of Xiong'an New Area is still in its initial stages, with limited urban expansion. The region's agriculture is relatively developed, and the focus on ecological conservation remains relatively stable with signs of improvement. Similar studies using remote sensing and ecological index (RSEI) have also yielded consistent conclusions in the past, which is confirmed through the analysis of land cover changes

from 2004 to 2015 that the intensity of area variations for impervious surfaces, vegetation, and water bodies is less than 5% in all cases [7].

The results of grey correlation analysis showed the correlation between precipitation and FVC was the most prominent. Hence, local vegetation displayed greater sensitivity to natural factors than human activities during this period. Nevertheless, it is worth noting that human activities can be a more prominent driving force in certain contexts. The univariate linear regression results show some discrepancies compared to the outcomes obtained from the grey relational analysis. But it is important to note that all $R^2$ values are less than one, with the values for urbanization rate and annual precipitation being less than 0.1. This can be explained that, since multiple factors influence FVC, the direct relationship between FVC and each driving force cannot be determined solely based on the results of univariate linear regression due to an interaction effect and limited factors. On the other hand, 15 years could provide with a limit sample size in the time range of this work. Hence, these results warrant further investigation.

Our work primarily offers a visual interpretation and analysis of the spatiotemporal patterns of FVC and its potential driving forces in Xiong'an New Area during the specified time period. The results provide detailed and observation-based insights into the vegetation conditions. As a result, further in-depth research can be pursued. For instance, regarding the current work, analyzing the spatial distribution of each FVC transition and employing additional mathematical and statistical methods to quantify and explore data information could be beneficial. Additionally, considering the seasonal variations of FVC, quantifying the relationship between FVC and land use, and forecasting the spatiotemporal changes of FVC in the future are potential avenues for future research.

All in all, it is reasonable to infer from the study's findings that there is significant room for improvement in the local socio-economic development level during the study period. As a newly planned district, Xiong'an possesses substantial development potential and educational value. Additionally, considering the strong correlation between precipitation and FVC, it can be deduced that precipitation holds predictive significance for local vegetation cover. Despite our understanding of the recent patterns of FVC in Xiong'an New Area, further investigation is required to explore the underlying relationship between such phenomena and regional development. Therefore, our next step will be to examine whether the current ecological environment of the new area aligns with the goals set in the development plan.

## 5. Conclusions

We conducted an analysis of the spatial-temporal variation characteristics of FVC in Xiong'an from 2005 to 2019 during the initial stages of Xiong'an's planning and development. The analysis utilized Landsat remote sensing images and employed the pixel dichotic model, transition matrix, and pixel-by-pixel spatial transfer. Moreover, we examined the correlation between FVC and various factors using a grey system approach. The main conclusions are:

- The FVC in Xiong'an exhibits significant fluctuations spatially, indicating an overall unstable ecological structure due to areas of lower FVC land such as cropland and grass for they are more susceptible to change.
- From 2005–2019, significant degradation of high-coverage vegetation has been accompanied by an increase in bare land. Although recent years have shown overall improvement, high-level degradation remains apparent especially in recent years.
- The Baiyangdian wetland has experienced an increase in water bodies and vegetation, indicating relative stability and slight improvement in FVC within this area.
- The grey correlation coefficient between precipitation and FVC is 0.67, indicating a positive correlation and establishing precipitation as the only driving force significantly associated with FVC in this study. Natural factors exhibit an average correlation coefficient of 0.69, whereas human factors show an average correlation coefficient of 0.60.

**Author Contributions:** Conceptualization, Z.H. and A.G.; methodology, Z.H., Y.C. and A.G.; software, Z.H.; validation, Z.H., H.Q. and J.W.; formal analysis, Z.H., Y.C. and J.W.; investigation, Z.H. and A.G.; resources H.Q.; data curation, Z.H., Y.C. and J.W.; writing—original draft preparation, Z.H., A.G. and H.Q.; writing—review and editing, Z.H., A.G. and H.Q.; visualization, Z.H., Y.C. and J.W.; supervision, H.Q.; project administration, A.G. and H.Q.; funding acquisition, A.G., H.Q. and Y.C. All authors have read and agreed to the published version of the manuscript.

**Funding:** This research was funded by the National Key Research and Development Program of China (Grant No. 2019YFE01277002).

**Institutional Review Board Statement:** Not applicable.

**Informed Consent Statement:** Not applicable.

**Data Availability Statement:** Our research data are from relevant open data websites, which can be obtained according to the links listed in Section 2.2 of this paper.

**Acknowledgments:** The authors would like to thank the high-performance computing support from the Center for Geodata and Analysis, Faculty of Geographical Science, Beijing Normal University. We are also very grateful to the anonymous reviewers for their valuable comments and suggestions for the improvement of this paper.

**Conflicts of Interest:** The authors declare no conflict of interest. The funders had no role in the design of the study, in the collection, analyses, or interpretation of data, in the writing of the manuscript, or in the decision to publish the results.

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
