# Peer review of "Spatial-Temporal Pattern and Driving Forces of Fractional Vegetation Coverage in Xiong’an New Area of China from 2005 to 2019"

_sustainability, doi:10.3390/su151511985_

Round 1
Reviewer 1 Report
Dear All,
Please, see posted comments.

Author Response
Dear reviewer,
We would like to thank you for your careful reading, helpful comments, and constructive suggestions, which has significantly improved the presentation of our manuscript.
We have carefully considered all comments and revised our manuscript accordingly. The manuscript has also been double-checked, and the typos and grammar errors we found have been corrected. In the attachment file, we summarize our responses to the comments. We hope our revised manuscript can be accepted for publication.
The detailed point-by-point response can be found in the attachment file.
Thanks again for your valuable advice.
Best regards,
All authors

Reviewer 2 Report
in the title, delete either pattern or variation....retain any of the words
Research gap should be mentioned clearly as it has to be focused in the introduction.
the significance or importance of research conducting in this area is to be mentioned
literature survey is not conducted extensively
materials and methods are nicely elucidated and presented lucidly
the results obtained are well presented and discussed. however, the interpretation at several instances is not consistent and not agreement with that observed by several earlier researchers.
for example, the interpretation written for figure 6 is not up to the mark and an extensive re investigation is required in this section.
even other figures such as 9, 10- the discussion should be elaborated. the discussion is very concise and limited as what is observed from the figure.
but the discussion is lack of scientific findings and scientific discussions based on the observations.
the authors should focus attention on the interpretation, scientific discussions, and consistency at several instances of paper.
occasionally, it is observed grammatical errors and typo errors too. hence should be corrected before submitting the revised version of manuscript.
Minor editing of English language is required and the same is mentioned in my comments to the authors.
Author Response

(The authors gave the same response as above.)

Reviewer 3 Report
The introduction presents information related to the methods used to analyse changes in vegetation. Still, it requires a more solid justification of the current methods for studying spatial time series data (using satellite images).
L109. It is mentioned that: "average annual precipitation is not specified in the given information". However, additional sources of information should be reviewed to complete this information as it is one of the driving forces. There is also no mention of the source of the climate data and the general characteristics of the study site.
Table 1 has no information on the source of the collected data and how to consult it. Likewise, the description of each of the data used is poor and does not allow an adequate follow-up of the development of the work. For example, the location of the weather station used is not specified. It is unclear what the period used for the study is; each data source has a different period without explaining the reason.
It is suggested to support and justify the reasons for using only three Landsat images in 2005-2019 to assess land use changes, considering that Landsat has a long historical series of data available. In addition, the work could be complemented with images such as Sentinel with higher spatial and temporal resolutions.
The description of the image processing could be improved by adding more details about the different processes used. For example, it is stated that radiometric calibration and atmospheric correction were carried out with ENVI software. Although it is mentioned that the FLASSH method was used, no references support this.
L298. The 2018 image is mentioned, but the 2019 image was used.
L366-376. The spatial changes in the vegetation cover from 2005 to 2019 are described, but the magnitude and possible causes are not mentioned. Its interpretation could be improved if a graph of frequencies of each land use by year is added.
L380. Reference is made to the NDWI index, but its calculation is not described, nor is its use substantiated in the methodology section.
Figure 4 could be improved by adding an FVC map with each land cover level, allowing a better understanding of the changes in the FVC.
For example, L482-483 says, "The conversion rates from medium to low cover vegetation into bare land were 24.57% and 35.29%, respectively,..." but the results correspond to the Mid-Low and Low classes. Similar cases are presented in the presented results. Assigning other class names to facilitate their interpretation and avoid confusion is suggested.
Changing the order of presentation of results is suggested, starting with the oldest period, 2005-2013 and ending with the entire period, 2005-2019.
In addition, it is recommended to detail which activities are referred to as "local ecological protection efforts" (L419-421) that had specific results in one area but could be improved in other sites where degradation (moderate and severe) is observed.
The interpretation of the results presented in Tables 4, 5 and 6 is limited to describing the values obtained. Although it is mentioned in some cases that there are no significant changes (e.g. L529), there is no evidence of this analysis.
The results of the grey relational analysis are presented in Table 7 but are not described or discussed in depth when this could be the work's main contribution.
The results presented in Figures 9b to 9d are the same as those shown in Figure 9a and are not described or contrasted with similar work.
The study area was planned to meet the objective of harmonising "green practices with high efficiency", so it would be interesting to know whether the results obtained in 2019 align with the goals set out in the original project.
The discussion is limited to the presentation and description of results, but there is no contrast with similar works and no explanation of the meaning of the results obtained and of the limitations; therefore, the work's substantial contribution is unclear.
Author Response

(The authors gave the same response as above.)

Round 2
Reviewer 3 Report
The corrected version is a significant improvement from the original version. However, some aspects can still be improved, described below.
The justification for using three LANDSAT images over such a long period is not robust and needs improvement. The main reason for using Landsat images in this type of study is precisely their long historical period of imagery, so they could be used to obtain more information related to the driving forces of land use change. If the justification is based on redundancy, it should be supported by similar work; otherwise, it is not supported.
The grammatical errors and typo errors should be corrected before submitting the new revised version of the manuscript. e.g. L375 says SOPT and should read SPOT, L 344, 345. It says Normalized Differential Water Body Index but should read Normalized Difference Water Index.
Figures 3, 5, 6, 7 and 8 have errors in the information presented. Some colours within the maps need to be correctly displayed.
The discussion is limited to the presentation and description of the results. Although it is mentioned that there was an improvement and contrasted with similar works, there is no evidence of that. The limitations must be described to clarify the work's substantial contribution.
Author Response

(The authors gave the same response as above.)
